# ECG Recordings as Predictors of Very Early Autism Likelihood: A Machine Learning Approach

**DOI:** 10.3390/bioengineering10070827

**Published:** 2023-07-11

**Authors:** Deepa Tilwani, Jessica Bradshaw, Amit Sheth, Christian O’Reilly

**Affiliations:** 1Artificial Intelligence Institute, University of South Carolina, Columbia, SC 29208, USA; amit@sc.edu (A.S.); christian.oreilly@sc.edu (C.O.); 2Department of Computer Science and Engineering, University of South Carolina, Columbia, SC 29208, USA; 3Carolina Autism and Neurodevelopment Research Center, University of South Carolina, Columbia, SC 29208, USA; jbradshaw@sc.edu; 4Institute for Mind and Brain, University of South Carolina, Columbia, SC 29208, USA; 5Department of Psychology, University of South Carolina, Columbia, SC 29208, USA

**Keywords:** autism spectrum disorder, biomarker, electrocardiograms, machine learning, heart rate variability

## Abstract

In recent years, there has been a rise in the prevalence of autism spectrum disorder (ASD). The diagnosis of ASD requires behavioral observation and standardized testing completed by highly trained experts. Early intervention for ASD can begin as early as 1–2 years of age, but ASD diagnoses are not typically made until ages 2–5 years, thus delaying the start of intervention. There is an urgent need for non-invasive biomarkers to detect ASD in infancy. While previous research using physiological recordings has focused on brain-based biomarkers of ASD, this study investigated the potential of electrocardiogram (ECG) recordings as an ASD biomarker in 3–6-month-old infants. We recorded the heart activity of infants at typical and elevated familial likelihood for ASD during naturalistic interactions with objects and caregivers. After obtaining the ECG signals, features such as heart rate variability (HRV) and sympathetic and parasympathetic activities were extracted. Then we evaluated the effectiveness of multiple machine learning classifiers for classifying ASD likelihood. Our findings support our hypothesis that infant ECG signals contain important information about ASD familial likelihood. Amongthe various machine learning algorithms tested, KNN performed best according to sensitivity (0.70 ± 0.117), F1-score (0.689 ± 0.124), precision (0.717 ± 0.128), accuracy (0.70 ± 0.117, *p*-value = 0.02), and ROC (0.686 ± 0.122, *p*-value = 0.06). These results suggest that ECG signals contain relevant information about the likelihood of an infant developing ASD. Future studies should consider the potential of information contained in ECG, and other indices of autonomic control, for the development of biomarkers of ASD in infancy.

## 1. Introduction

Autism Spectrum Disorder (ASD) is a neurodevelopmental disorder characterized by social communication and interaction challenges, restricted interests, and repetitive behaviors [1]. Studies have shown that early intervention can enhance developmental outcomes in individuals with ASD, with earlier onset in infancy leading to better outcomes [2]. In the United States, the prevalence of ASD has been increasing and is currently estimated to affect approximately 1 in 36 children [3], representing about 2.3% of the population. Research indicates that having a family history of ASD increases the likelihood of developing the condition [4], with studies indicating a higher risk among those with siblings, cousins, or parents diagnosed with ASD [5,6]. This statistic underscores the need for developing early and low-cost ASD indicators that could support earlier, more efficient diagnoses, personalized treatments, and improved quality of life for people with ASD. Furthermore, earlier diagnosis of ASD could lead to the development of even earlier parent-implemented behavioral interventions that support very early communication and social interaction skills [7].

Widely accepted assessment tools for ASD include the Autism Diagnostic Observation Schedule (ADOS) [8,9] and the Autism Diagnostic Interview (ADI) [8,10]. These measures offer many advantages. First, when conducted by trained clinicians, they inform reliable diagnoses and allow for a more systematic and reproducible behavioral characterization of participants across studies. Second, ADI and ADOS profiles enable the formation of ASD subgroups based on perceived or confirmed symptoms that may be associated with various clinical trajectories and genetic markers [8]. However, these standardized assessments of ASD require advanced professional training and a more comprehensive evaluation than many medical conditions.

Biosignals are crucial in understanding and monitoring neurodevelopmental disorders. These signals, derived from various physiological processes such as cognitive processes, muscular activations, or cardiac activity, provide valuable insights into the functioning of the human body. By analyzing biosignals, such as electroencephalograms (EEG), electromyograms (EMG), and electrocardiograms (ECG), healthcare professionals can diagnose and monitor disorders with greater accuracy. Biosignals enable the detection of abnormalities, patterns, and trends that may indicate the presence or progression of a disorder or the response to treatment. They also facilitate personalized treatment plans, allowing healthcare providers to tailor interventions based on individual biomarker profiles. These can potentially be used to derive reliable diagnostic biomarkers of ASD. Researchers are increasingly studying EEG to monitor abnormal brain development in clinical settings. For example, one study used spectral and nonlinear features of EEG signals to predict the diagnosis of ASD in infancy [11]. However, collecting high-quality EEG data in infants or young participants is subject to substantial methodological hurdles contingent upon the experimental settings and the intended analyses [12]. A whole-brain functional magnetic resonance imaging (fMRI) study [13] of children with ASD and children with developmental language disorders revealed decreased functional connectivity between the frontal and parietal-occipital regions and between the anterior and posterior insular cortex and the limbic cortex. In addition, due to common anatomical brain regions involved in both autonomic dysfunction and social-emotional dysregulation [14,15], measures of autonomic activity constitute a prime candidate for the development of a biomarker and for the study of ASD in general.

Previous research indicates that ASD symptoms could be linked with sympathetic arousal and parasympathetic activity [16,17]. Electrodermal activity (EDA) is one of the primary measures of sympathetic arousal. In [18], the authors discovered a significant connection between anxiety symptoms and EDA in children with ASD. Children with ASD and high anxiety levels demonstrated substantially reduced EDA relative to ASD participants with low anxiety and neurotypical controls [19]. To the best of our knowledge, no one has studied the use of extracted features from ECG with parasympathetic and sympathetic symptoms in individuals with ASD, despite the potential benefits of such an approach.

In recent decades, some studies have identified features extracted from electrocardiogram (ECG) signals associated with ASD [20,21]. Among these promising ECG features is heart rate variability (HRV), as it can serve as a potential non-invasive biomarker. In addition, HRV reflects alterations in parasympathetic and sympathetic nervous system activity, and lower HRV in ASD may reflect abnormal neuronal connectivity within the autonomic nervous system (ANS) [21,22]. Various metrics are utilized to quantify HRV features, aiming to extract biomarkers that hold physiological significance. Multiple preclinical and clinical studies have been conducted on these metrics and have proven effective in identifying pathophysiological states in different clinical situations [23,24,25]. Some studies have conducted comparisons of HRV patterns between children with ASD and control participants [26,27,28]. These studies have yielded several findings related to HRV, one of which is that individuals with ASD exhibit significantly lower HRV during social stress. Second, specifically in terms of the parasympathetic-specific index known as RSA (respiratory sinus arrhythmia), individuals with ASD demonstrate significantly lower RSA and RSA reactivity during social stress, social debriefing, and cognitive tasks when compared to the control group.

Recently, researchers have proposed many Machine Learning (ML) approaches to improve the early identification of ASD and to study potential ASD biomarkers with MRI [29,30,31]. However, utilizing MRI may not be viable for infants between the ages of 3–6 months. Infants at this age are typically very active and may not be able to stay still for an extended period, which can result in poor image quality. Earlier contributions used classical ML methods (such as with MRI/fMRI [29,32,33,34,35]). More recently, deep learning methods have demonstrated a considerable advantage over classical approaches due to their ability to rely on hidden representations of extracted features in MRI/fMRI [36]. Other non-invasive methods have been coupled with ML for ASD. Eye tracking is a good example. It has proven to be a valuable tool, with or without deep learning, in investigating ASD by capturing and analyzing gaze patterns and attentional mechanisms [37,38]. These studies have provided insights into the distinct eye movement patterns observed in individuals with ASD, including reduced eye contact, a preference for non-social stimuli, and difficulties in facial recognition and joint attention. However, it is important to acknowledge the limitations of eye tracking in ASD research. One limitation is the inherent variability in eye movement patterns among individuals with ASD, making establishing consistent norms and benchmarks challenging. Technological limitations, such as difficulties in calibration and participant compliance, can also affect the quality and reliability of the collected data. Furthermore, researchers may need to improve the ecological validity of their laboratory-based eye-tracking studies to capture the full range of social interactions that people experience in real-world settings. Although an increasing number of studies proposed ML methods for ASD diagnosis using behavioral or psychophysiological data such as EEG [39,40], scant attention has been given to the application of ML to ECG signals for predicting ASD. The integration of HRV measures and sympathetic and parasympathetic analysis will contribute to our understanding of autonomic dysregulation in individuals diagnosed with ASD. This investigation is made important by the tight relationship between autonomic control and attention regulation, since a breakdown in such regulatory processes at a very early age may play a role in the development of this disorder [41]. In this study, we aimed to demonstrate that ECG contains valuable predictive information for ASD prognostic and further motivate such investigation on the relationships between ASD and the autonomic system, as measured through ECG. Toward that goal, we used non-invasive ECG sensors to record infants’ heart activity during interactions with toys and their caregivers. Participants were infants with a typical or elevated likelihood of developing ASD based on familial history. We pre-processed the ECG signals and extracted HRV features to train our ML models. By investigating how predictive these models are, we aim to evaluate the potential of ECG-captured autonomic activity to indicate ASD likelihood.

## 2. Methods

### 2.1. Acquisition Protocol

A small ECG sensor (Actiwave Cardio; CamNTech) with a pair of electrodes was placed on the chest of the infants to record raw signals. The infants then participated in two experiments: one involving interactions with objects only (OIX) and the second involving interactions with their caregiver (PIX). During the OIX condition, caregivers placed infants on their laps, and five objects were presented sequentially to infants on the table for 1–2 min each. The caregiver did not interact with their infant during this time. During the PIX condition, infants were lying on their backs on a table, and caregivers interacted with them without toys for nine minutes. Both experiments were conducted with the same infants. Only OIX or PIX recording is available for some participants, either because of technical issues or because the infant’s fussing required early experiment termination. Recordings from N = 70 infants of age 3–6 months were available for this study. Participants were either at elevated familial likelihood for ASD (EL) because they had an older full biological sibling with ASD (n = 31) or at a typical likelihood for ASD (TL) given the absence of a family history of ASD in first or second-degree relatives (n = 39).

### 2.2. ECG Pre-Processing

Due to changes in hardware during the data collection, ECG signal collection was conducted at three different frequencies, i.e., 128 Hz, 512 Hz, and 1024 Hz. We processed the signals at their original frequency. Various artifacts can contaminate ECG, including power line interference, channel noise, electromyographic artifacts, electrode contact noise, and baseline wandering due to body movement and respiration. Since such artifacts harm automatic classification, we cleaned the ECG signals using the Python HeartPy library [42].

We used a custom procedure to determine missing beats (i.e., heartbeats that HeartPy failed to detect) and fix the time series of interbeat intervals (i.e., conceptually, imputing the timing of R peaks for missing beats). This procedure goes as follows. We first divided the time series of interbeat intervals x(n) by its median value to obtain $\hat{x(n)}=x\left(n\right)/median\left(x\left(n\right)\right)$. Each value in $\hat{x(n)}$ indicates how long the detected interbeat interval is in the number of median interbeat intervals. Then, we rounded $\hat{x(n)}$ to the closest integer such that $\tilde{x(n)}=round\left(\hat{x(n)}\right)$. We identified $\tilde{x(n)}-1$ missing beats for the n-th interval. To pinpoint the exact timing of these missing beats, we assigned each beat an index starting at zero and incrementing by one for each beat (including the identified missing beats). We used this index $\tilde{n}$ as the new independent variable for linear interpolation. For example, using the arguments as defined in the NumPy interp(x, xp, fp) function, we implemented this interpolation as $x\left(\tilde{n}\right)=interp\left(\tilde{n},n,x\left(n\right)\right)$.

Following this procedure, we obtained time series of interbeat intervals corrected for missing beats. To avoid biasing our analysis with recordings that were too noisy for reliable R peak extraction, we excluded recordings with more than 30% of interpolated peaks. Furthermore, to ensure enough peaks for reliable feature estimation, we rejected segments with less than 30 beats. Examples of noisy segments removed because of high noise are shown in Figure 1A,B. The ECG signals corrected for baseline wandering are shown in Figure 1C. Figure 1D illustrates power line interference or muscle response, which is corrected during pre-processing. Figure 1E gives an example of a clean signal following pre-processing. Figure 2 shows an example of zoomed-in clean R peaks. We performed this pre-processing on different segment lengths (30 s, 60 s, 90 s, 120 s, and full-length) of ECG data to test the effect of such a segmentation. ECG data is divided into segments using non-overlapping sliding windows of different lengths (30, 60, 90, or 120 s). By examining various segment lengths, we can assess whether shorter or longer segments are more effective in accurately identifying and classifying ECG signals, regardless of the individual heartbeats or specific characteristics of ECG waves [43]. In addition, this approach allows for assessing the impact of segment duration on the system without considering individual heartbeats or particular characteristics of ECG waves. Since the recording length can impact the success of the inclusion criteria aforementioned, different epoch lengths result in different sample sizes (30 s: N = 61, 60 s: N = 54, 90 s: N = 56, 120 s: N = 57, full-length: N = 51). Figure 3a illustrates the automated ECG pre-processing process as explained.

### 2.3. Feature Extraction and Preliminary Analysis

After pre-processing, the process takes the average of the features extracted from each segment within a recording. For instance, if we divided a recording into five 90-s segments, we averaged the features across these five segments. We employed a Linear Mixed Effects Model (“feature condition + Labels + Month”) to analyze the impact of a particular feature while controlling for other variables, including age (measured in months), condition (PIX/OIX), and labels (LL/EL). There was not a statistically significant impact of age on the feature values (p’s > 0.1), and so repeated within-subject recordings were aggregated by taking the average of the segment-averaged vectors across the different recording sessions (i.e., at different ages). For example, if we have recordings at three, four, and six months for a given participant, each consisting of 90 s averaged vectors, we took the average of these vectors to obtain a single averaged vector for this participant across the three time points. It is important to note that this process generates a matrix dependent on the segments’ length. For each segment length, we have different sets of subjects (due to pre-processing criteria) and ten HRV features per subject. These features were extracted (Table 1) using the Python neurokit2 library [44]. We provided the clean peaks as an array of ECG data points to calculate HRV features (MeanNN, MaxNN, MinNN, MedianNN, pnn20, CVNN, sd1sd2) based on normal beat intervals (NN). NN is the interval between consecutive R peaks (Figure 2). Further, we calculated the Cardiac Sympathetic Index (CSI) and the Cardiovagal Index (CVI) as proxies for sympathetic and parasympathetic activation and their neuronal and cardiac dependencies. To calculate CSI and CVI, the variability of R peaks is observed and transformed into an elliptic distribution using Lorenz plots [45]. The length of the longitudinal (L) and transverse (T) axes is calculated within each elliptic distribution. We then obtain CVI = log10(L*T) and CSI = L/T. After extracting the features, we calculated the correlation coefficient between each feature (Figure 4) and rejected redundant features if their correlation coefficient was greater than 0.90. The final set of included features was: MedianNN, pnn20, CVNN, sd1sd2, HTI, CSI, and CVI.

### 2.4. EL vs. TL Classification Using Machine Learning

We used Scikit-learn [50] to test various classifiers (Gradient Boosting (GB), Extra Trees (ET), Random Forest (RF), Ada Boost (AB), Decision Tree (DT), K Nearest Neighbors (KNN), XGBoost (XGB), and Multi-layer Perceptron (MLP)) to classify infants into EL and TL groups. We used the nested cross-validation to systematically evaluate the performance of different combinations of hyperparameters for each model to classify infants into EL and TL groups. Cross validation is explained in Section 2.5. For the final evaluation, we used 100 stratified shuffles with an (80/20)% split. We reported our findings using the mean and standard deviation (std) from the distribution of 100 stratified splits. Finally, we extracted feature importance. The importance of each feature is calculated based on how much they contribute to the model’s predictions. We utilized the permutation importance function (in Scikit-learn [51]) with the ROC metric for calculating feature importance, which is model-agnostic. This function assesses a feature’s significance by randomly changing its values and measuring the model’s resulting decrease in performance. The permutation importance function enables us to gauge a feature’s importance in predicting outcomes. Shuffling values randomly will cause an important feature to a more significant decrease in performance compared to a less important one.

### 2.5. Cross Validation

To evaluate the classification performance of our models, we implement the nested cross-validation technique. We use two nested loops of cross-validation. The outer loop divides the dataset into 100 stratified shuffle splits, where each split separates a test set (20%) for final evaluation from the remainder of the data (80%) used by the inner loop. This outer loop helps estimate the model’s performance on unseen data. Within each outer loop iteration, the inner loop performs another round of cross-validation. It further divides the remaining data into multiple folds of training (64%) and validation subsets (16%) using a grid search approach. The inner loop is responsible for hyperparameter tuning by systematically trying out different combinations of hyperparameters and evaluating their performance on the validation sets. A nested cross-validation approach allows for a fair comparison and selection of the best hyperparameters. This process ensures that performance is evaluated on completely unseen data and helps prevent overfitting and information leakage. Figure 3b illustrates our nested cross-validation approach.

### 2.6. Analysis Pipeline

The ECG sensors were well tolerated by all infants. Overall, we rejected 12% recordings (9 participants). Figure 3 illustrates our methodology for determining ASD genetic likelihood using ECG signals. It includes automated ECG pre-processing with the following steps: filtering, segmentation of ECG data, pre-processing to eliminate artifacts, extraction of clean R peaks, missing peaks correction, feature extraction, and classification. After pre-processing, the features described in Section 2.3 were extracted. For each participant, we computed the average of the extracted features from the remaining segments across all age time points. These features are used for our ML models as input.

## 3. Results

In this study, we conducted a comparative analysis of eight classification algorithms (ADB, DT, ET, GB, KNN, MLP, RF, and XGB) to identify the most predictive model for distinguishing between EL and TL groups. We employed the following segment lengths of ECG to evaluate the efficacy of shorter versus longer segments in accurately classifying these groups: 30, 60, 90, and 120 s and the full length. We used accuracy, precision, sensitivity, specificity, and F1-score to evaluate the performance of these models. The definitions of these metrics can be found in [52]. We also calculated the area under the receiver operating characteristic curve (ROC), a widely used metric for evaluating classification model performance [53]. This metric is less affected by class imbalance than accuracy [54]. Figure 5 shows heatmaps comparing the performance of different machine learning models (y-axis) on the given task across the five selected evaluation metrics (panel) and different segment lengths (x-axis). These heatmaps were color-coded to visually depict the performance levels, with brighter colors indicating better performance and darker colors indicating poorer performance. In our study, we utilized nested cross-validation, as explained in Section 2.5, to reliably assess the performance of our various models without risking overfitting or information leakage. Through this evaluation, we determined that tree-like models outperformed other models. We observed diverse performance outcomes across the evaluation metrics among these tree-based models, namely, KNN (kd-tree), DT, and the ADB model. In Table 2, we show the results (mean ± std) obtained using nested cross-validation of these models. These models generally exhibit similar trends across different time lengths for accuracy. We conclude that KNN is a top performer. The distributions of accuracy (0.70 ± 0.117; *p*-value: 0.02) and area under the ROC curve (0.686 ± 0.122; *p*-value: 0.06) obtained through bootstrapping with shuffle splits statistically support that this classifier performs above chance level. Further, this model displayed the highest sensitivity, scoring 0.70 ± 0.117, indicating its ability to identify positive instances correctly. KNN also achieved an impressive F1-score of 0.689 ± 0.124, reflecting its balanced precision and recall. With a precision score of 0.717 ± 0.128, KNN demonstrated its capability to minimize false positives. KNN and ADB consistently achieve the highest accuracy, with values ranging from 0.600 to 0.700, followed by DT, XGB, and GB. However, MLP consistently performs the poorest in terms of accuracy. The precision performance shows a similar pattern, with KNN and ADB achieving the highest precision scores across all segment lengths. The performance of the DT model also stands out in terms of precision. The ET, RF, GB, and XGB models achieve similar precision scores, while MLP consistently lags. When considering sensitivity, the models generally display similar trends in accuracy and precision, with KNN, DT, and ADB showing the best performances. MLP again demonstrates the lowest sensitivity across all segment lengths. Analyzing the F1 score, KNN performs well consistently, followed by DT, ADB, and XGB, while MLP falls behind. Finally, regarding ROC performance, KNN consistently achieves the highest scores, closely followed by DT and ADB. MLP demonstrates the poorest ROC performance throughout the different segment lengths. KNN, DT, and ADB consistently demonstrate strong performance across multiple evaluation metrics, making them potentially favorable models for classifying EL and TL.

In Table 3, we have shown the results from two other studies [47,55] that employed ML classifiers. Specifically, our study demonstrated enhanced precision, recall, and accuracy compared to these prior works. In Figure 6, we identified that “MedianNN” is the best feature based on feature importance from the KNN model. Feature importance is shown in percentage in Figure 6 on the x-axis and features on the y-axis, suggesting that these features do not have significant discriminatory power when distinguishing between EL and LL classes. Though all other features have proven highly effective in achieving improved classification results, further investigation is needed to determine if additional factors or considerations could make them more meaningful for the classification task. From Table 2, we observe that KNN has stable results across all five metrics across the 90 s segments. We also see the differences between the EL and LL groups in their ECG 90 s segment extracted features, indicating changes in the ANS of EL infants. A thorough comparison of the extracted ECG features between the EL and LL groups is presented in Table 4. The findings indicate that the EL has comparatively lower HRV features compared to the LL. This can be associated with a reduced ability of the ANS to adapt and respond to different physiological and environmental demands.

## 4. Discussion

The primary goal of this study was to determine whether infant ECG signals help to classify ASD likelihood. Statistically, EL infants show elevated rates of ASD diagnosis and elevated ASD symptoms, making this preliminary study a first step in developing biomarkers for early ASD diagnosis and better understanding the relationships between autonomic regulation and ASD. Our approach has high ecological validity, considering our naturalistic experimental paradigm (self-initiated, naturalistic interactions with objects and parents) and the wearable, wireless, and non-invasive sensors. In addition, such an approach can be easily scaled up and reach more families by collecting data at home or remotely (i.e., without an on-site experimenter). Traditional ECG pre-processing techniques have several constraints, including their incapability to scale up for remote set-ups, proneness to human errors, limitations in handling noise, and difficulties in standardization. As a result, there is a growing interest in automated ECG pre-processing algorithms, which have the potential to enhance the accuracy, efficiency, and standardization of ECG signal analysis in clinical and research settings. Our pipeline is illustrated in Figure 3a. This explains an automated algorithm that detects and removes artifacts in the ECG signal, applies filters to remove noise and unwanted components, segments the signal into relevant segments, and improves overall signal quality. Automated pre-processing can save time and increase efficiency in ECG signal analysis by reducing the need for manual inspection and annotation of the signal. We utilized the neurokit [44] library to extract features automatically, eliminating the need for manual feature crafting. This approach proved successful in analyzing our dataset. We chose not to use the very popular Convolutional Neural Network (CNN) approach for this work for a few reasons. First, CNN may not reveal how heart rate variability (HRV) changes over time. CNNs learn from ECG signals when provided as input with extensive training data. However, our dataset is small, and CNN would not generalize well [56]. Additionally, we aimed to report changes in the data on a group basis. A CNN would not be suitable for this analysis since it is difficult to trace how the network arrives at its conclusions or determines the features that drive its predictions. Using feature extraction methods, we could analyze the data and observe group-wise changes more effectively. Moreover, this approach provides transparency and interpretability in understanding the underlying patterns and variations within different groups, which could not be accomplished via a CNN.

Several studies suggest a significant correlation between social functioning and sympathetic and parasympathetic markers, which can be assessed through HRV, in typical development [57,58,59]. This has led to a growing interest in exploring their association with ASD. By studying HRV features, we may better understand how ANS function is related to the core and associated deficits of ASD and other diagnoses related to neurodevelopmental disorders. In addition, understanding ANS could lead to a clearer understanding of the underlying mechanisms contributing to the development and manifestation of these conditions. Initially, we selected ten features as described in Section 2.3 for our study. However, we excluded MaxNN, MinNN, and MeanNN because of high correlations between these features and MedianNN, leaving seven features. The median is a measure of central tendency less sensitive to extreme values or outliers than the mean.

We aimed to evaluate whether the overall HRV features could separate the EL group from the TL group. Research shows a link between lower HRV and atypical brain functioning in individuals with ASD. Therefore, our study hypothesized that differences in autonomic regulation specific to ASD would result in HRV differences between groups. To achieve this, we calculated two variables, CSI and CVI, associated with various psychological and physical health outcomes, including social behavior, cognition, and emotional functioning. Our results showed that CSI values were higher and CVI values were lower in EL infants compared to TL infants, indicating lower-paced breathing and a lower heart rate in EL infants. Additionally, EL infants exhibited lower median, pNN20, HTI, and CVNN values than TL infants. These findings suggest a potential relationship between ASD and altered ANS functioning, particularly regarding parasympathetic regulation. These values relate to a previous study that found that children and adults with ASD have lower HRV than TL individuals, indicating decreased parasympathetic activity and/or increased sympathetic activity [60]. This autonomic dysregulation has been associated with various behavioral and physiological problems in individuals with ASD, including social difficulties, anxiety, and gastrointestinal issues. For reference, Table 4 shows the mean ± std of HRV features for each group in the 90 s segment length. We used nested cross-validation. We employed the 100-stratified shuffle-splitting technique on outer loop to ensure a more reliable assessment of the finalized models and minimize any bias. We calculated the average outcome across all 100 splits to report our findings. Despite the acknowledged limitation of a relatively small dataset, cross-validation was advantageous as it provided a more robust evaluation of the models. It repeatedly split the data into training and testing subsets, allowing for a better performance assessment and reducing the potential for bias. Furthermore, it aided in estimating the generalization performance of the models on unseen data. We reported the results by taking a mean across all iterations of the 100 splits.

Our F1-score in KNN 0.689 ± 0.124suggests that wearable, wireless, and easy-to-use ECG devices hold potential for clinical applications in evaluating ANS activation and screening for ASD likelihood. By comparison, multi-electrode EEG devices are more sensitive to artifacts, making data collection with infants more complicated [61]. Other neuroimaging techniques, like fMRI, are even less practical due to higher costs and lower accessibility. While ECG signal disruption can occur, denoising and pre-processing methods can help recover some portion of noisy recordings.

Comparing the accuracy of our system with previously reported results on a very similar problem [47] (i.e., ASD diagnosis vs. ASD likelihood classification) suggests the improved performance of our approach. KNN is particularly well-suited for capturing complex decision boundaries thanks to its ability to consider the local structure of the data. This makes it highly effective when decision boundaries exhibit nonlinear or irregular characteristics. When applied to HRV features, it can effectively capture nonlinear relationships and interactions among features. KNN determines the class label by conducting a majority vote based on its neighboring data points. This inherent flexibility empowers KNN to deliver strong performance even when the data deviates from specific statistical assumptions. However, earlier studies [47] did not report important metrics like specificity, sensitivity, precision, or F1-score. This limits the extent to which we can make a fair comparison. Moreover, reporting only accuracy is known to be biased in the case of an unbalanced dataset (i.e., high accuracy can be obtained on an unbalanced dataset even if the classifier performs very poorly for the smaller class). Since reporting only accuracy is generally insufficient to assess the performance of a classifier properly, we also reported on precision, sensitivity, specificity, and F1-score. Further, instead of reporting point estimates, we reported these measures’ mean and standard deviation to allow future studies to test the statistical significance of performance differences. The results reported by the authors [55] show a higher ROC than what we found, which seems inconsistent with their reported accuracy. Further, the authors report multi-class classification performance for several ML algorithms using HRV for ASD, conduct problems, depression, and typical development. To our knowledge, these are the only other studies reporting classification results using ECG and HRV with ML for diagnostic applications in ASD. Moreover, these performances are only partly comparable (e.g., in [55], school-aged children were studied, while we reported on infants aged 3–6 months). The encouraging performances observed in our study support the use of HRV as a potential biomarker for monitoring the effectiveness of interventions for ASD and evaluating the physiological changes that occur in infants with ASD over time.

Some limitations need to be considered when interpreting the results of this study. Firstly, the study analyzed HRV parameters in a moderate sample size. Therefore, corroborating our conclusions in larger samples is essential to ensure the generalizability of the findings. In addition, this study balanced infants with an elevated familial likelihood for ASD and controls, resulting in a small proportion of infants with later ASD diagnoses. Thus, it is worth emphasizing that we did not aim to classify ASD diagnosis but the likelihood of ASD (or risk level). Our approach uses ECG and HRV to extract predictive information on autonomic control and its modulation by shifts in attention, a process associated with ASD by previous studies [41,48,49]. If not well regulated, this crucial function may have a developmental impact on other dimensions, including social communication, interaction with objects and people, and behavior patterns. In addition, a thorough assessment of the child’s history and functioning is necessary for an accurate diagnosis.

## 5. Conclusions and Future Work

ECG recordings are non-invasive, easy-to-use signals that can be leveraged in biomarker research and provide measures (e.g., heart rate variability) that index key neurological systems, including the sympathetic and parasympathetic nervous systems. This study found that we can use HRV features extracted from ECG to predict the familial likelihood of ASD in 3–6-month-old infants. In this study, we only used 10 HRV features. In the future, this study can be expanded along different axes including (1) exploring additional HRV features, including time-domain, frequency-domain, and nonlinear HRV features; and (2) replicating these findings in larger samples of infants and children, including those with confirmed ASD diagnoses and those with non-ASD developmental disorders such as language or attention deficit hyperactivity disorder.

## Figures and Tables

**Figure 1 bioengineering-10-00827-f001:**
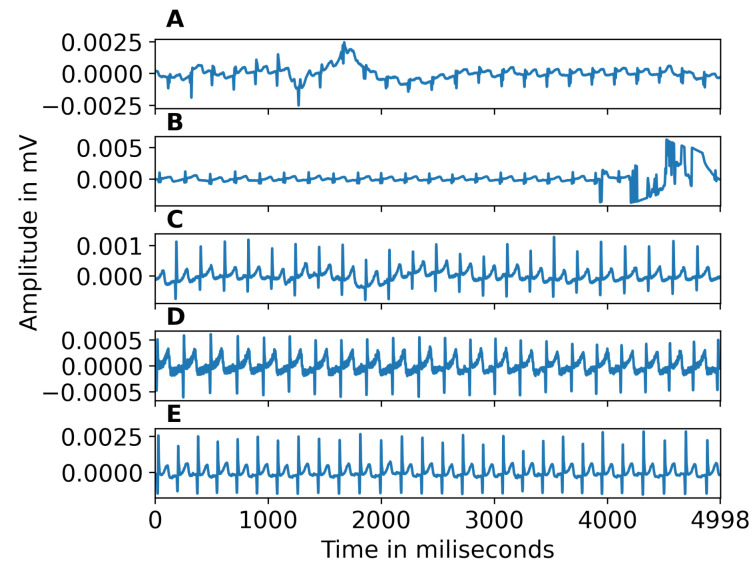
(**A**,**B**) Examples of segments excluded from the dataset because they were too noisy. (**C**) Example of baseline wandering. (**D**) Example of power line interference. (**E**) Clean signal, after pre-processing.

**Figure 2 bioengineering-10-00827-f002:**
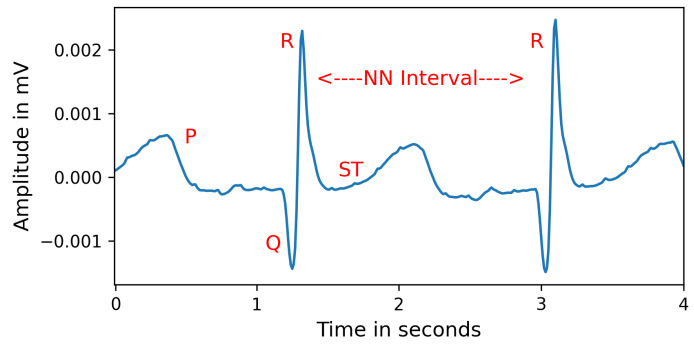
An example of zoomed-in cleaned ECG beats. NN stands for normal beat intervals, and is similar to “R-R intervals”, but highlights that these values represent normal cardiac timing, free from artifacts.

**Figure 3 bioengineering-10-00827-f003:**
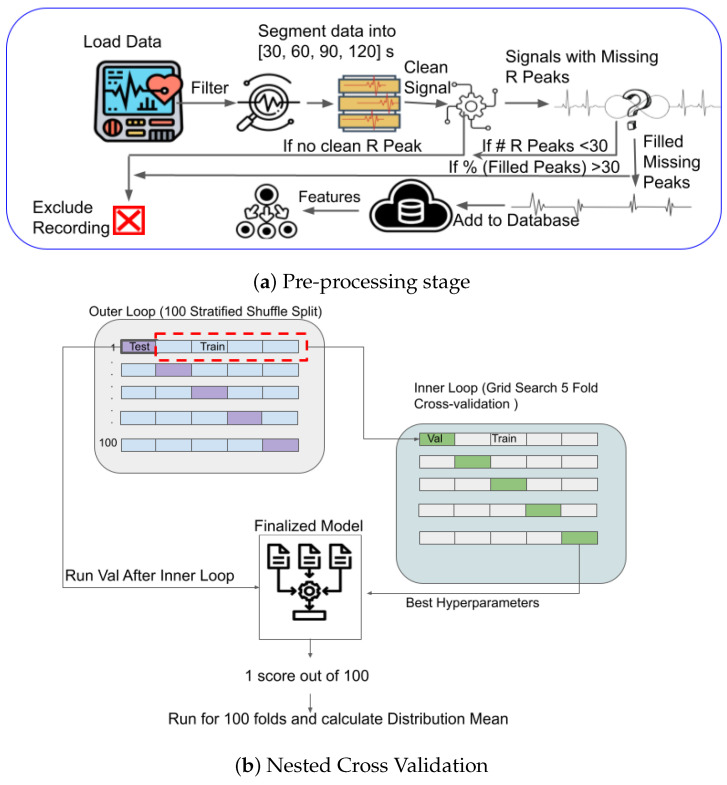
Schematic representation of the automated pipeline we used for ASD classification: (**a**) Shows pre-procesing as explained in Section 2.2, (**b**) Shows nested cross-validation we used for our approach.

**Figure 4 bioengineering-10-00827-f004:**
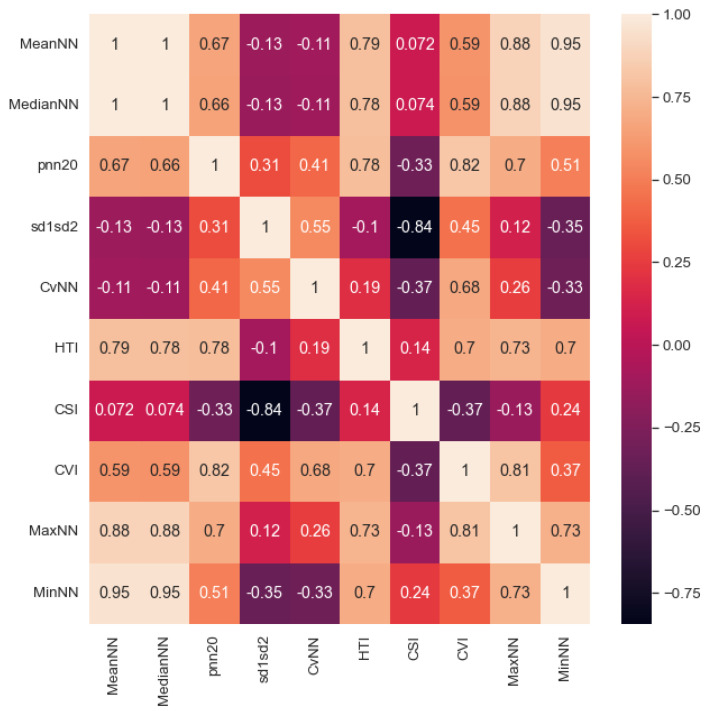
Heatmap of correlation between all the HRV features averaged across 90 s segment length.

**Figure 5 bioengineering-10-00827-f005:**
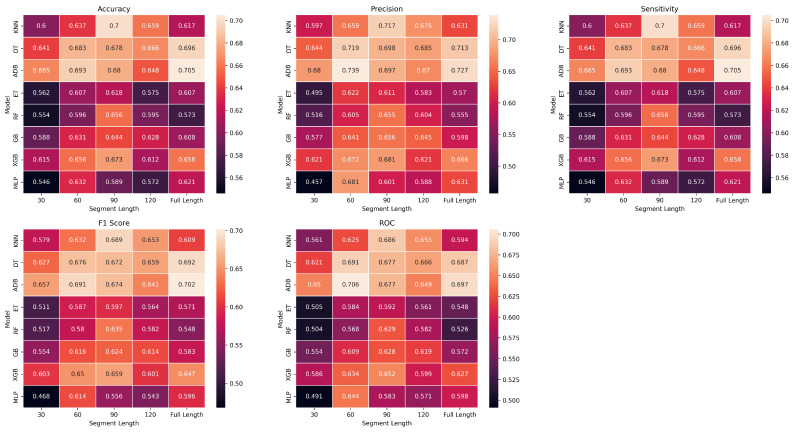
Heatmaps representing various metrics across different segment lengths (x-axis) for ML classifier experiments (y-axis). The heatmap provides a color-coded summary of the performance of each classifier for segment length, with darker colors indicating poorer results and lighter colors showing better results.

**Figure 6 bioengineering-10-00827-f006:**
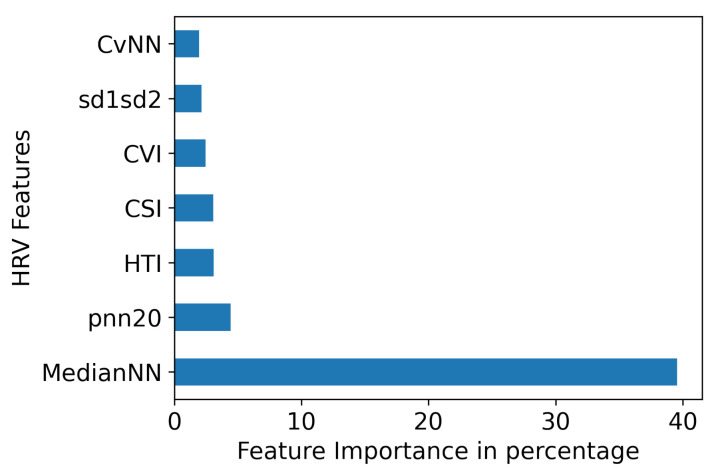
HRV Feature importance for KNN.

**Table 1 bioengineering-10-00827-t001:** Extracted features from ECG Signals after the pre-processing step.

Features	Feature Description
MeanNN	Mean of the NN [46,47].
MedianNN	Median of the NN [46,47].
pNN20	Proportion of successive NN with a difference larger than 20 ms. This measures the relative frequency of changes in heart rhythm [22].
sd1sd2	In a Poincaré plot, sd1 is standard deviation perpendicular to the line of identity and sd2 is standard deviation along the line of identity. sd1sd2 is the ratio of (sd1/sd2), an indicator of the unpredictability of the NN to measure autonomic balance when there is sympathetic activation [22].
CVNN	Coefficient of variation of NN, calculated as standard deviation of NN divided by mean of NN [26].
HTI	HRV Triangular Index is the integral of the NN density histrogram divided by its height [22].
CSI	The CSI quantifies the sympathetic nervous system activity indicating increased arousal [48].
CVI	The CVI quantifies the parasympathetic nervous system activity [49].
MaxNN	Max of the NN [46,47].
MinNN	Min of NN [46,47].

**Table 2 bioengineering-10-00827-t002:** Top Three Models with All Segment Lengths and Metrics’ (mean ± std).

ADB							
Segment Length	Accuracy p-val	Accuracy (Mean ± std)	ROC p-val	ROC (Mean ± std)	Sensitivity (Mean ± std)	Precision (Mean ± std)	F1 (Mean ± std)
30	0.07	0.665 ± 0.111	0.12	0.65 ± 0.124	0.665 ± 0.111	0.68 ± 0.128	0.657 ± 0.116
60	0.06	0.693 ± 0.136	0.07	0.706 ± 0.139	0.693 ± 0.136	0.739 ± 0.135	0.691 ± 0.141
90	0.06	0.68 ± 0.128	0.1	0.677 ± 0.135	0.68 ± 0.128	0.697 ± 0.14	0.674 ± 0.132
120	0.1	0.648 ± 0.129	0.14	0.649 ± 0.133	0.648 ± 0.129	0.67 ± 0.139	0.641 ± 0.13
Full Length	0.04	0.705 ± 0.116	0.07	0.697 ± 0.129	0.705 ± 0.116	0.727 ± 0.127	0.702 ± 0.119
**KNN**							
**Segment** **Length**	**Accuracy** **p-val**	**Accuracy** **(Mean ± std)**	**ROC** **p-val**	**ROC** **(Mean ± std)**	**Sensitivity** **(Mean ± std)**	**Precision** **(Mean ± std)**	**F1** **(Mean ± std)**
30	0.19	0.60 ± 0.108	0.27	0.561 ± 0.108	0.600 ± 0.108	0.597 ± 0.132	0.579 ± 0.111
60	0.13	0.637 ± 0.114	0.19	0.625 ± 0.125	0.637 ± 0.114	0.659 ± 0.129	0.632 ± 0.117
90	0.02	0.70 ± 0.117	0.06	0.686 ± 0.122	0.70 ± 0.117	0.717 ± 0.128	0.689 ± 0.124
120	0.08	0.659 ± 0.13	0.09	0.655 ± 0.13	0.659 ± 0.13	0.675 ± 0.133	0.653 ± 0.133
Full Length	0.17	0.617 ± 0.126	0.24	0.594 ± 0.136	0.617 ± 0.126	0.675 ± 0.133	0.653 ± 0.133
**DT**							
**Segment** **Length**	**Accuracy** **p-val**	**Accuracy** **(Mean ± std)**	**ROC** **p-val**	**ROC** **(Mean ± std)**	**Sensitivity** **(Mean ± std)**	**Precision** **(Mean ± std)**	**F1** **(Mean ± std)**
30	0.11	0.641 ± 0.116	0.16	0.621 ± 0.132	0.641 ± 0.116	0.644 ± 0.145	0.627 ± 0.125
60	0.06	0.683 ± 0.133	0.1	0.691 ± 0.144	0.683 ± 0.133	0.719 ± 0.158	0.676 ± 0.144
90	0.04	0.670 ± 0.121	0.09	0.677 ± 0.125	0.678 ± 0.121	0.698 ± 0.136	0.672 ± 0.126
120	0.05	0.666 ± 0.127	0.13	0.666 ± 0.131	0.666 ± 0.127	0.685 ± 0.139	0.659 ± 0.133
Full Length	0.11	0.696 ± 0.137	0.15	0.687 ± 0.154	0.696 ± 0.137	0.713 ± 0.157	0.692 ± 0.143

**Table 3 bioengineering-10-00827-t003:** The performance results of models documented in the literature (mean ± std).

Methodology	Precision	Sensitivity	F1-Score	ROC	Accuracy
Ensemble [47]	-	-	-	-	0.75
					(avg. accuracy)
XGB (their highest performing model) [55]	0.57 ± 0.22	0.59 ± 0.14	-	0.88 ± 0.12	0.59 ± 0.22

**Table 4 bioengineering-10-00827-t004:** Mean ± Std values of the HRV features extracted averaged across 90 s segments of ECG.

Feature	MeanNN	HTI
**TL**	571.33 ± 214.77	5.64 ± 2.38
**EL**	419.47 ± 175.86	5.01 ± 2.02
**Feature**	**MedianNN**	**CSI**
**TL**	569.13 ± 213.75	3.38 ± 1.23
**EL**	417.09 ± 176.22	3.56 ± 1.61
**Feature**	**pnn20**	**CVI**
**TL**	11.7 ± 11.2	3.67 ± 0.52
**EL**	8.75 ± 11.86	3.57 ± 0.58
**Feature**	**sd1sd2**	**MaxNN**
**TL**	0.35 ± 0.15	704.71 ± 313.01
**EL**	0.39 ± 0.27	520.11 ± 225.89
**Feature**	**CvNN**	**MinNN**
**TL**	0.04 ± 0.02	501.48 ± 198.36
**EL**	0.06 ± 0.03	347.18 ± 149.89

## Data Availability

Data are not shared at this time because the data collection is still ongoing. For reproducing the analyses, we made our code available on https://github.com/lina-usc/ECG-ASD-Likelihood-Classification acessed on 21 June 2023.

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
