# Peer review of "ECG Recordings as Predictors of Very Early Autism Likelihood: A Machine Learning Approach"

_bioengineering, 2023, doi:10.3390/bioengineering10070827_

Round 1

Reviewer 1 Report

The authors focused on utilizing ECG signals for Infants to predict Autosim besides the family history. I have several comments on the presented work:

1. The physiological background describes the relationship between ECG and Autosim.

2. The authors should declare the importance of biosignals in human disorders. 

3. The authors have to clarify why they choose ECG, not EEG. And there are many studies focused on EEG signals to show that.

4. Is there a relationship between family history and autism?

5. Is the less activity of infants with the surrounding sufficient to judge the Atosim?

6. I think the authors have to clear in detail the pre-processing technques for ECG signal

7. Did you segment the signal to identify the period between two successive QRSs?

Over all the paper need work.

Reviewer 2 Report

Interesting work, I would like to thank the authors for this contribution. The study presents an interesting ML application of in the autism context. Additionally, I commend the authors for acknowledging potential limitations. However, I would like to offer some suggestions for improvement to the next version, please.

(1)

It is unclear if cross-validation was used to split the dataset, despite acknowledging its relatively small size as a limitation. Utilizing cross-validation would have been beneficial due to the dataset's size, allowing for more reliable model evaluation and mitigating biases.

(2)

I have concerns regarding the dataset splitting process, particularly in relation to the potential occurrence of "information leakage." Specifically, I am concerned about the inclusion of samples from the same participant in both the train and test sets. This scenario would result in information leakage, compromising the validity of the evaluation. I kindly request clarification on how you addressed this issue and ensured the prevention of information leakage during the dataset splitting process.

(3)

The process of feature extraction is well-described, but it is worth discussing why the study did not consider using Convolutional Neural Networks (CNNs) to automate this process. Hand-crafted features are becoming less common, and CNNs offer advantages such as automatic learning of meaningful features from raw data, capturing hierarchical representations, and eliminating the need for manual feature engineering.

(4)

The literature review demonstrates a commendable effort. However, it would be valuable to expand the discussion on non-invasive methods coupled with machine learning (ML) for autism diagnosis, such as eye-tracking. Including eye-tracking as an example would enrich the related work section by encompassing additional modalities that have shown promise in the field. One possible example to mention: https://doi.org/10.2196/27706

(5)

The Scikit-Learn reference should be cited properly, please.

Pedregosa, F., Varoquaux, G., Gramfort, A., Michel, V., Thirion, B., Grisel, O., ... & Duchesnay, E. (2011). Scikit-learn: Machine learning in Python.  Journal of Machine Learning Research12, 2825-2830.

Overall, I appreciate the authors' efforts and look forward to seeing an improved version of this study.

Round 2

Reviewer 1 Report

I think the authors should add more clarification about the results

Reviewer 2 Report

Thank you for addressing my previous comments. However, I still have concerns regarding the issue of 'information leakage' mentioned in my second point. Since this could be a significant factor influencing the study's results, I remain skeptical about their validity.

Generally good.
